# Dynamics of Pressure Variation in Closed Vessel Explosions of Diluted Fuel/Oxidant Mixtures

Venera Giurcan, Domnina Razus , Maria Mitu  and Codina Movileanu *

"Ilie Murgulescu" Institute of Physical Chemistry, Romanian Academy, 202 Spl., Independentei, 060021 Bucharest, Romania
* Correspondence: cmovileanu@icf.ro

**Abstract:** Nitrous oxide is widely used as oxidizer or nitriding agent in numerous industrial activities such as production of adipic acid and caprolactam and even for production of some semiconductors. Further, it is used as an additive in order to increase the power output of engines, and as an oxidizer in propulsion systems of rockets, because it has a large heat of formation ($+81.6$ kJ mol$^{-1}$). $N_2O$ is highly exothermic, and during its decomposition a supplementary heat amount is released, so it needs special handling conditions. The combustion of fuels in nitrous oxide atmosphere can lead to high unstable and turbulent deflagrations that speedily self-accelerate and therefore a deflagration can change to a detonation. The peak explosion pressure and the maximum rate of pressure rise of explosions in confined spaces are key safety parameters to evaluate the hazard of processes running in closed vessels and for design of enclosures able to withstand explosions or of their vents used as relief devices. The present study reports some major explosion parameters such as the maximum (peak) explosion pressures $p_{max}$, explosion times $\theta_{max}$, maximum rates of pressure rise $(dp/dt)_{max}$ and severity factors $K_G$ for ethylene-nitrous oxide mixtures (lean and stoichiometric) diluted with various amounts of $N_2$, at various initial pressures ($p_0 = 0.50$–$1.50$ bar), in experiments performed in a spherical vessel centrally ignited by inductive-capacitive electric sparks. The influence of the initial pressure and composition on $p_{max}$, $\theta_{max}$ and $(dp/dt)_{max}$ is discussed. The data are compared with similar values referring to ethylene-air mixtures measured in the same initial conditions. It was found that at identical C/O ratios with ethylene-air, ethylene-$N_2O$-$N_2$ mixtures develop higher explosion pressures and higher rates of pressure rise, due to the exothermic dissociation of $N_2O$ under flame conditions.

**Keywords:** ethylene; nitrous oxide; nitrogen dilution; pressure dynamics; confined explosions

## 1. Introduction

Combustion is the major energy production process, so the study of flammability features of gaseous combustible mixtures at various initial conditions still represents a major challenge for researchers. Fuel–air combustion has been intensively studied, but in recent years an important concern has been raised: not only to find alternative fuels to substitute the conventional fuels, but also to find alternative oxidizers to replace oxygen. Nitrous oxide ($N_2O$, also known as laughing gas) is a strong oxidizing agent, roughly equivalent to hydrogen peroxide, and much stronger than oxygen gas. It was used as oxidizer in the semiconductors industry and as a rocket propulsion system [1–3]. On the other hand, $N_2O$ is highly exothermic and during its decomposition a supplementary heat amount is released, so its handling is critical.

As an oxidizer, $N_2O$ can easily decompose to oxygen and nitrogen at elevated pressures and temperatures [2,4]. Numerous studies were published on fuel-$N_2O$ mixtures in terms of flammability parameters such as flammability limits, ignition energy, flame speeds, etc. The explosion ignition and propagation in $H_2$-$CH_4$-$NH_3$-$N_2O$-$O_2$-$N_2$ mixtures was studied by Pfahl et al. [5]. They reported flammability limits, ignition energies and flame speeds

in combustion vessels at initial pressure of 1 bar and initial temperature of 295 K. Some investigations on $N_2O$ flames focused on the flammability domain of different fuel-$N_2O$ mixtures (flammability limits) in the presence of inert gases [6,7].

The explosion parameters of toluene/$N_2O$ at initial pressure of 1 or 6 bar and initial temperature of 343 K from measurements in an 8 L spherical vessel were reported by Vandebroek et al. [6]. The explosion features of volatile organic vapours, such as n-pentane, diethyl ether and diethyl amine, and of $C_1$-$C_8$ n-alkanes in $N_2O$ atmosphere in a cylindrical explosion vessel were studied by Koshiba et al. [2,8] and compared with those of mixtures containing $O_2$.

Vandooren et al. [9] conducted a study on the flame structure of stoichiometric methane-$N_2O$-Ar and methane-$O_2$-Ar by means of the molecular beam sampling and mass spectroscopy methods. The propagation parameters and laminar burning velocities of $CH_4$-$N_2O$ in the presence of various inert gases (e.g., He, Ar, $N_2$ and $CO_2$) obtained from experiments in a 0.52 L spherical vessel, at initial pressures from 0.3 to 1.8 bar were recently investigated by Razus et al. [10,11]. The authors reported that the maximum explosion pressure as well as the maximum rates of pressure rise present a linear dependence on the initial pressure. The additive efficiency on the explosion parameters decreased starting with $CO_2$, followed by $N_2$, Ar and He. The overall activation energy of methane oxidation by $N_2O$, which is lower than the overall activation energy when $O_2$ from air is used as an oxidant, can be due to $N_2O$ dissociation.

Ethylene-$N_2O$ flame was less studied despite the safety concerns raised from handling and storage of nuclear waste, where these mixtures can form, as well as due to their use in aerospace applications. Ethylene is a high-reactivity fuel used in the chemical industry, but previous investigations focused on explosion of ethylene-air (oxygen) mixtures [12–16]. However, ethylene–$N_2O$ mixtures have recently been proposed as a promising propellant for hydrazine replacement [17,18] due to their advantages: similar vapor pressures, which allow a good miscibility, and high vapor pressure, which allows a self-pressurizing without an external pressure supply. Additionally, ethylene–$N_2O$ mixtures are not carcinogenic, have a relatively low cost and a low toxicity and after their combustion, no ammonia results in comparison with hydrazine. Therefore, ethylene–$N_2O$ mixtures represent a green propellant and thus, the studies related to these mixtures are always necessary.

Shen et al. [19] experimentally investigated the explosion dynamics of inert-diluted $C_2H_4$-$N_2O$ explosions in a standard 20-L spherical vessel at sub-atmospheric pressure (inerts: $N_2$ and $CO_2$). For inert concentrations lower than 30% they found a correspondence between the maximum explosion pressure experimentally measured and the corresponding Chapman–Jouguet (CJ) detonation pressure, meaning the activation of a DDT (Deflagration-to-Detonation) mechanism during the explosion process, also demonstrated by Bane et al. [20] for nitrogen-diluted hydrogen–nitrous oxide flames. The explosion characteristics of $C_2H_4$-$N_2O$-$N_2$ mixtures measured in various cylinders (0.17 L, 1.18 L and 2.0 L), at different initial pressures, were reported by Movileanu et al. [21]. They found a linear dependence of the maximum explosion pressure (as well as of the maximum rise of pressure rise) on the initial pressure, similar to that reported for fuel-air mixtures [16]. In smaller volume explosion vessels, no DDT occurred. The DDT in long tubes, the explosion limits, and the ignition delay times of explosions in $C_2H_4$-$N_2O$ and $C_2H_4$-$O_2$ mixtures were also investigated in the previous studies [22–26].

From the above literature review it can be concluded that researchers have focused their interest mostly on studying the explosion characteristics of other fuel–$N_2O$ flames rather than that of ethylene-$N_2O$ flames. Therefore, these latter flames must be studied in detail to cover the gap that exists regarding them in literature.

In the present study, the dynamics of closed-vessel explosion of nitrogen-diluted $C_2H_4$/$N_2O$ mixtures is examined by means of the maximum (peak) explosion pressures, explosion times, maximum rate of pressure rise and severity factors. Lean and stoichiometric $C_2H_4$/$N_2O$ mixtures diluted with various nitrogen amounts were examined using a 0.52 L vessel with spherical aspect and central ignition at different initial pressures. The nitrogen

was chosen as diluent because it is frequently used as an inert gas in the case of explosion inhibition. The data and conclusions from the present research could contribute to a better understanding of explosions occurring in $N_2O$ atmosphere, providing new insights of these processes.

## 2. Materials and Methods

The experimental system consists of a 0.52 L stainless steel spherical vessel, the ignition device, the data acquisition system and a vacuum line. A schematic representation of the experimental set-up is presented in Figure 1.

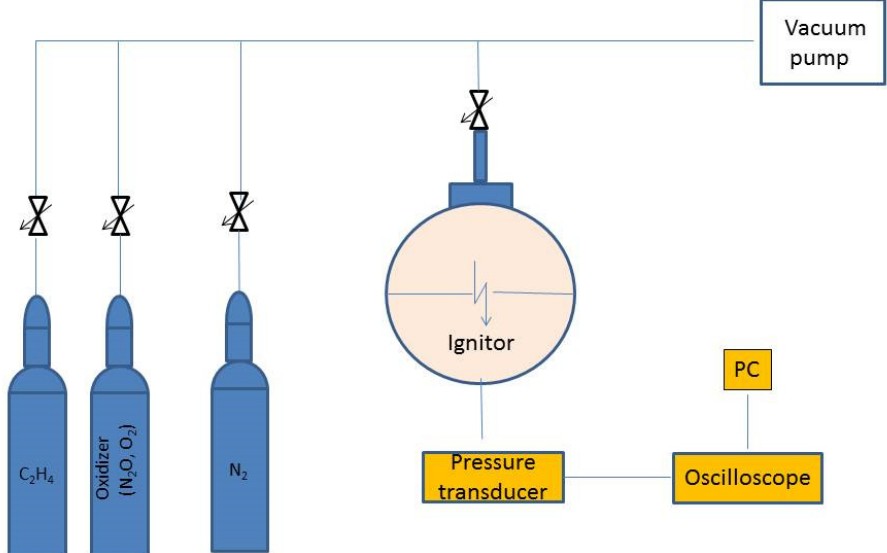

**Figure 1.** Experimental system.

The explosion vessel has several ports for the gas feed/evacuation valve, pressure transducers, ionization probes, and ignition electrodes. The explosion vessel was tested for static pressures of minimum 40 bar. The ignition, centrally located inside the vessel, was done using inductive-capacitive sparks of high voltage. A piezoelectric pressure transducer (Kistler 601A), in line with a Charge Amplifier Kistler 5001 SN is used for monitoring the pressure evolution. The signals provided by the ionization probe and pressure transducer were acquired using a Tektronix digital oscilloscope type TDS 2014B connected to a PC, usually at 2500 signals/channel and maximum 1 GS/s sampling rate. Details regarding the experimental setup and procedure can be found in other published works [16,27,28]. Before each experiment, the experimental vessel was evacuated to a pressure down to $5 \times 10^{-4}$ bar, then the explosive mixture was admitted and allowed 3 min. to become quiescent. After that, the gaseous mixture was ignited and then evacuated.

The $C_2H_4$–$N_2O$–$N_2$ gaseous mixtures were prepared in a metallic cylinder using the partial pressure method at a total pressure of 4 bar. Each investigated mixture was used 24 h after mixing their components. For every examined mixture, the equivalence ratio of the $C_2H_4$–$N_2O$ mixture was given, together with the added nitrogen amount, as the volumetric $N_2$ concentration in the end mixture ($C_2H_4$ + $N_2O$ + $N_2$).

Ethylene (99.99% from SIAD, Bergamo, Italy), $N_2O$ and $N_2$ (both 99.999% from SIAD, Bergamo, Italy) were used, without any purification. Ethylene–$N_2O$ mixtures having equivalence ratios from 0.8 to 1.0, diluted by 40–60% $N_2$, were studied. For concerns of laboratory operation safety, $C_2H_4$–$N_2O$–$N_2$ explosions were examined at total initial pressures from 0.5 to 1.50 bar. For each initial condition of the explosive mixture, a minimum of 3 experiments were performed (using fresh $C_2H_4$–$N_2O$–$N_2$ mixture each time) to minimize the operational errors. At the end, the obtained results were averaged.

### 3. Computing Programs and Data Evaluation

The 0-D COSILAB package [29] allowed the calculation of the adiabatic explosion pressures and adiabatic flame temperatures. This program is based on an algorithm that allows the calculation of the equilibrium composition of the products of a fuel–oxidizer gaseous mixture according to the thermodynamic criterion of chemical equilibrium: the minimum of Gibbs energy (keeping constant the temperature and pressure) or minimum of Helmholtz energy (keeping constant the temperature and volume). As combustion products we considered a total of fifty-three compounds. The runs were performed for lean and stoichiometric ethylene-$N_2O$ mixtures, various nitrogen concentrations (40 to 60 vol.%), at ambient initial temperature and at different initial pressures (0.50–1.50 bar).

The $p$(t) curves were smoothed by Savitzky–Golay method in order to obtain the rates of pressure rise. Utilisation of the Savitzky–Golay method means the analysis of a number of points (e.g., 500–700 points) registered from the ignition to the time necessary to reach the maximum explosion pressure and provide a smoothed first derivative without filtering the data. This method implies the least squares quartic polynomial fitting across a moving window within the data. A 10% smoothing level was used for each curve, since a higher value of smoothing level (e.g., 20%) leads to a reduction of the noise and to a signal distortion.

### 4. Results

An example of pressure-time record for stoichiometric $C_2H_4$-$N_2O$-$N_2$ mixture explosion in spherical vessel with central ignition at 1 bar initial pressure and ambient initial temperature is presented in Figure 2 and compared with that obtained for stoichiometric $C_2H_4$-air explosion registered under the same initial conditions [30]. The comparison can be performed since the examined mixtures have close flame temperatures.

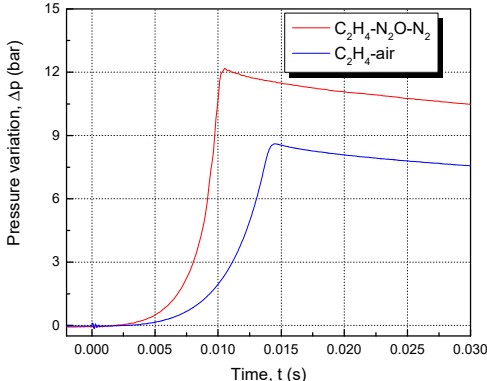

**Figure 2.** Comparison of pressure-time records for stoichiometric $C_2H_4$-$N_2O$-$N_2$ (60 vol.% $N_2$) and stoichiometric $C_2H_4$-air [30] explosions in a spherical vessel having central ignition at $p_0$ = 1 bar and $T_0$ = 298 K.

It can be observed that for equivalent initial conditions the use of $N_2O$ for replacing $O_2$ as oxidizer entails the increase of the maximum explosion pressure and the decrease of the explosion time. This shows us that the fuel–$N_2O$ mixtures are more reactive when compared with the mixtures with air with the same flame temperature. This behaviour is due to the additional heat amount released from the exothermal decomposition of $N_2O$ when $N_2O$ is used as oxidizer instead of air ($O_2$).

For all pressure–time trajectories, the pressure first increases progressively, and then decreases once the maximum values are reached, this fact indicating that a complete combustion has taken place.

The pressure evolutions of $C_2H_4$-$N_2O$-$N_2$ explosions at initial pressure of 1 bar and various amounts of nitrogen are presented in Figure 3 as examples. The comparison presented in this figure shows that, for the same initial pressure, temperature and fuel:oxidizer ratio, the explosion pressure decreases when the inert concentration increases. It can

be seen that, starting with 40 vol.% dilution, the pressure inside the explosion vessel rises up smoothly after mixture ignition. This behaviour has been widely reported by other researchers [16,31,32].

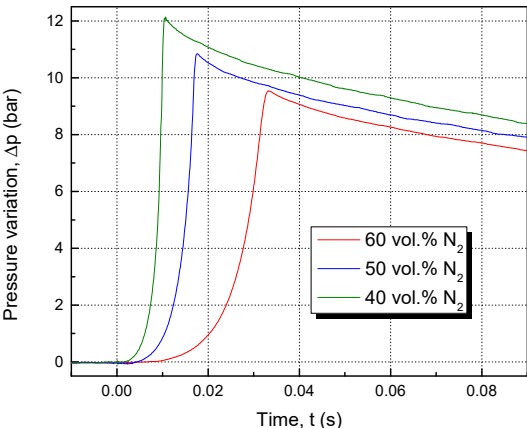

**Figure 3.** Pressure trajectories of stoichiometric $C_2H_4/N_2O/N_2$ mixtures at $p_0 = 1$ bar.

The initial pressure is one of the parameters that influence the maximum (peak) explosion pressure. An example of maximum (peak) explosion pressures' variation against initial pressures for stoichiometric $C_2H_4$-$N_2O$-$N_2$ mixtures is presented in Figure 4 for various $N_2$ dilution ratios and ambient initial temperature. Such variations were also observed for lean $C_2H_4$-$N_2O$-$N_2$ mixtures. As expected, the maximum explosion pressure during the explosion of $C_2H_4$-$N_2O$-$N_2$ mixtures increases as the initial pressure increases.

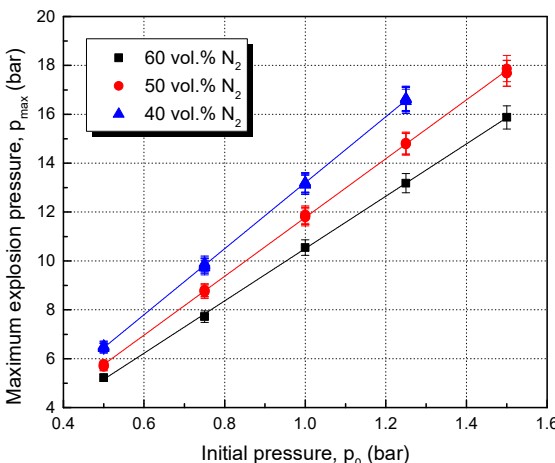

**Figure 4.** Maximum explosion pressures of stoichiometric $C_2H_4$-$N_2O$-$N_2$ at various initial pressures.

Due to the lack of experimental data reported in literature for explosions of $C_2H_4$-$N_2O$-$N_2$ mixtures in spherical vessels at initial pressures $p_0 \geq 1$ bar, we can only compare our data with those reported at sub-atmospheric pressures. Our data registered at $p_0 = 0.5$ bar are: $p_{max} = 6.50$ bar for stoichiometric $C_2H_4$-$N_2O$ mixture diluted with 40 vol.% $N_2$ and $p_{max} = 5.70$ bar for stoichiometric $C_2H_4$-$N_2O$ mixture diluted with 50 vol.% $N_2$. These results are higher than those reported by Shen et al. [19] for mixtures with the same composition and initial pressure obtained from experiments in a 20 L spherical vessel ($p_{max} = 5.70$ bar for the stoichiometric $C_2H_4$-$N_2O$ mixture diluted with 40 vol.% $N_2$ and $p_{max} = 5.00$ bar for stoichiometric $C_2H_4$-$N_2O$ mixture diluted with 50 vol.% $N_2$). These discrepancies might be due to the differences in vessel volumes, which can affect the heat losses and through them, the explosion pressures.

Linear correlations of maximum explosion pressures with initial pressures of the mixture were found for all examined systems. The correlation could be described using the following equation:

$$p_{max} = a + b \cdot p_0 \qquad (1)$$

The coefficients $a$ and $b$ from Equation (1) are necessary to find the unknown values of the explosion pressures at a certain value of the initial pressure in the studied pressure range without experimental measurements, as long as the combustion is propagating as a deflagration. They depend on a series of parameters such as: initial temperature, nature of the mixture, vessel shape, and aspect ratio. The coefficients of linear regressions for stoichiometric and lean $C_2H_4$-$N_2O$-$N_2$ mixtures are presented in Table 1.

**Table 1.** Slope and intercept of $p_{max}$ vs. $p_0$ linear regressions.

| $\varphi$ | $[N_2]$ (vol.%) | $-a$ (bar) | $b$ | $r_n$ |
|---|---|---|---|---|
| | 60 | $0.198 \pm 0.103$ | $10.70 \pm 0.09$ | 0.999 |
| 1.0 | 50 | $0.252 \pm 0.055$ | $12.03 \pm 0.05$ | 0.999 |
| | 40 | $0.298 \pm 0.040$ | $13.50 \pm 0.04$ | 0.999 |
| | 60 | $0.332 \pm 0.065$ | $10.11 \pm 0.06$ | 0.999 |
| 0.8 | 50 | $0.334 \pm 0.125$ | $11.52 \pm 0.12$ | 0.999 |
| | 40 | $0.342 \pm 0.066$ | $12.98 \pm 0.07$ | 0.999 |

Such linear correlations between explosion pressures and initial pressures were observed not only for fuel–air mixtures [16], but also when other fuel–$N_2O$ inert mixtures were examined [10].

Besides the initial pressure, the shape and size of the explosion vessel influence the explosion pressure. Thus, in cylindrical vessels, the explosion pressures are smaller in comparison with those registered in spherical vessels due to higher heat losses that occur in these cylindrical enclosures. On the other hand, the higher vessel size is, the smaller the explosion pressure is. The influence of the size and aspect of the explosion vessels on this parameter can be observed from Table 2, where data referring to stoichiometric $C_2H_4$-$N_2O$-$N_2$ diluted with 60 vol.% $N_2$ measured in a cylinder with L/D = 1.0 and in a cylinder with L/D = 1.5 measured at initial pressures between 0.50 and 1.50 bar collected from [21] are given along with the present data. All data from Table 2 refer to experiments with central ignition.

**Table 2.** Influence of vessel shape and size on the maximum explosion pressures of $C_2H_4$-$N_2O$ mixtures diluted with 60% $N_2$.

| Vessel | Volume (L) | $p_0$ (bar) | $p_{max}$ (bar) | Reference |
|---|---|---|---|---|
| Spherical | 0.52 | 0.50 | 5.22 | Present data |
| | | 0.75 | 7.72 | |
| | | 1.00 | 10.54 | |
| | | 1.25 | 13.18 | |
| | | 1.50 | 15.87 | |
| Cylindrical L/D = 1.0 | 0.17 | 0.50 | 3.94 | [21] |
| | | 0.75 | 4.51 | |
| | | 1.00 | 6.36 | |
| | | 1.25 | 8.10 | |
| | | 1.50 | - | |
| Cylindrical L/D = 1.5 | 1.18 | 0.50 | - | [21] |
| | | 0.75 | 5.00 | |
| | | 1.00 | 6.84 | |
| | | 1.25 | 8.89 | |
| | | 1.50 | 10.75 | |

The presence of an inert diluent affects the maximum explosion pressures as can be seen in Figure 5, where data referring to both studied mixtures (lean and stoichiometric) at ambient initial conditions are given. As expected, the higher the nitrogen amount is, the smaller the explosion pressure is. Adding an inert (such as nitrogen) to a gaseous flammable

mixture leads to an alteration of both thermodynamics and kinetics of the combustion process, so that a higher inert concentration leads to a small explosion pressure.

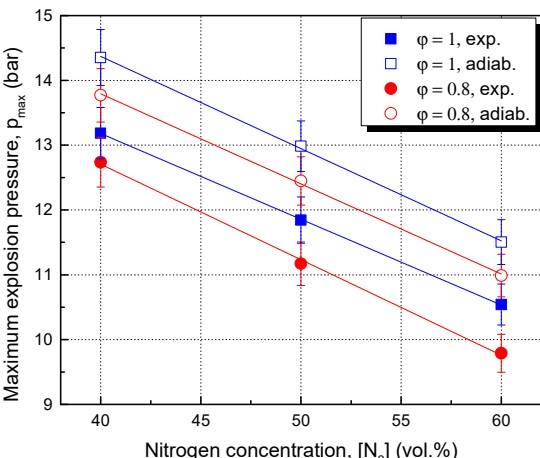

**Figure 5.** Maximum explosion pressure as a function of inert concentrations at ambient initial temperature and pressure.

The kinetic effect due to $N_2$ addition on $C_2H_4$-$N_2O$ mixtures can be assigned mainly to $N_2$ participation in the following reaction:

$$O + N_2 (+M) \rightarrow N_2O (+M) \tag{R1}$$

The study undertaken by Shen et al. [19] pointed out that, at high initial pressures, nitrogen acts as both as a reactant and a third body, and thus reaction (R1) becomes important. Therefore, although the $N_2O$ undergoes an energetic decomposition through the reaction (R2) written below, nitrogen can effectively inhibit $N_2O$ decomposition, as described in [19].

$$N_2O (+M) \rightarrow N_2 + O (+M) \tag{R2}$$

Furthermore, the nitrogen addition can lead to a decrease in the branching ratio of the following reaction (R3) over reaction (R1), by its third-body collisional effect [19]:

$$H + O_2 \rightarrow O + OH \tag{R3}$$

Besides the kinetic effect, nitrogen addition contributes to the reduction of the flame temperature and flame speed, acting as a heat sink.

For all studied mixtures, a linear dependency of explosion pressure on inert concentration was observed. This dependency can be described using a correlation proposed by Oancea et al. [33,34] based on heat balance of the isochoric combustion of a fuel–air mixture in non-adiabatic conditions:

$$p_{max} = p_0 \left( \xi + \frac{r_l}{\nu_l} \cdot \frac{\Delta_c U'}{\overline{C_{e,V}} \cdot T_0} \right) - q_{tr} \cdot \frac{\gamma_e - 1}{V_0} \tag{2}$$

where $p_{max}$ represents the maximum explosion pressure attained at pressure $p_0$; $\xi = n_e/n_0$ is the ratio between the final ($n_e$) and initial ($n_0$) mole numbers; $r_l = n_l/n_0$ represents the ratio between the number of moles corresponding to the limiting component of the mixture and the total initial number of moles; $\nu_l$ represents the stoichiometric coefficient of the limiting component in the mixture; $\Delta_c U'$ represents the heat of combustion (at temperature $T_0$ and constant volume) corrected by taking into account the accompanying secondary reactions; $\overline{C_{e,V}}$ represents the molar heat capacity of the end gaseous mixture, averaged for the end components and for the temperature range $T_0$ to $T_{e,V}$; $q_{tr}$ represents the amount of heat

transferred by the gas to the vessel before the end of combustion; and $\gamma_e$ represents the adiabatic coefficient of the burned gas, at the end of combustion.

For the isochoric combustion of a gaseous fuel-oxidizer mixture, Equation (2) holds as long as it is assumed that the studied gas behaves like an ideal gas and that $p_{max}$ is approximately equal to the pressure at the combustion end. However, in the real cases, the gaseous fuels are made up from polyatomic molecules, having many degrees of freedom, and their heat capacities are high and depend on temperature, which leads to explosion pressures which differ from those obtained with this equation. Keeping constant the initial temperature, the maximum explosion pressures of $N_2$-diluted $C_2H_4$-$N_2O$ mixtures depend on initial pressures (influencing the molar heat capacity of the end gaseous mixture and the adiabatic coefficient of the burned gas, at the end of combustion) and on the amount of heat transferred by the gas to the vessel before the end of combustion. Therefore, we can observe that $N_2$ addition to $C_2H_4$-$N_2O$ mixtures influences both the slope and intercepts of $p_{max}$ versus $p_0$ plots.

Over a well-defined initial pressure domain, we can assume that $\xi$, $\Delta_c U'$and $\overline{C_{e,V}}$ do not depend on $p_0$ (or that their variations compensate each other) and therefore, the amount of heat transferred from the burned gas to the explosion vessel before the combustion ends, can be obtained using the intercepts of the linear correlations $p_{max}$ vs. $p_0$, whose values are presented in Table 1. Representative results regarding the amount of heat transferred by the stoichiometric and lean ethylene-$N_2O$-$N_2$ flames to the vessel's walls are presented in Table 3. In the present case, $q_{tr}$ was calculated following [16], as:

$$q_{tr} = -\frac{a \cdot V_0}{\gamma_e - 1} \tag{3}$$

**Table 3.** Transferred heat amount, $q_{tr}$, and fraction of heat lost, *F*, by stoichiometric and lean ethylene-$N_2O$-$N_2$ flames to vessel's walls.

| $N_2$ (vol.%) | $\gamma_e$ | | $q_{tr (J)}$ | | **F** | |
|---|---|---|---|---|---|---|
| | $\varphi = 1.0$ | $\varphi = 0.8$ | $\varphi = 1.0$ | $\varphi = 0.8$ | $\varphi = 1.0$ | $\varphi = 0.8$ |
| 40 | 1.3097 | 1.3139 | 0.333 | 0.550 | 0.020 | 0.036 |
| 50 | 1.3078 | 1.3119 | 0.426 | 0.557 | 0.023 | 0.032 |
| 60 | 1.3080 | 1.3121 | 0.503 | 0.570 | 0.024 | 0.029 |

From Table 3 it is seen that an increase of the nitrogen concentration leads to larger amounts of heat being lost to the vessel's wall due to the longer explosion times and lower propagation speeds for both lean and stoichiometric ethylene-$N_2O$-$N_2$ mixtures.

Another important parameter that describes the heat transferred during explosions in closed vessels is the fraction (*F*) of the heat transferred (lost) from the total released heat. This property depends, on one hand, on the corrected heat of combustion and the average heat capacity of burned gas (factors influenced by the composition of fuel and oxidizer) and, on the other hand, on the initial pressure of the studied mixture. In this study, the fractions *F* of transferred heat were calculated using the following equation [35], derived also from Equation (2):

$$F = \frac{-a}{p_0(b-1)} \tag{4}$$

Their values calculated at $p_0 = 1$ bar are also presented in Table 3. At constant inert composition, lower *F* values were obtained for the stoichiometric $C_2H_4$-$N_2O$ flames compared to the lean flames. This is due to the fact that the stoichiometric mixtures have shorter explosion times and higher burning velocities when compared to the lean mixtures.

When a certain amount of inert (molar fraction: $x_i$) is added to a fuel-air mixture, it is expected that the heat transferred from the burned gas to the vessel, before the combustion

ends, to be a function of mixture composition. Therefore, the term $q_{tr}$ from Equation (2) becomes $q_{tr}^* = f(r_l, x_i)$ and Equation (2) is rewritten as:

$$p_{max} = \left[ p_0 \left( \xi + \frac{r_l}{v_l} \cdot \frac{\Delta_c U'}{C_{e,V} \cdot T_0} \right) - q_{tr}^* \cdot \frac{\gamma_e}{V_0} \right] - \left( p_0 \frac{r_l}{v_l} \cdot \frac{\Delta_c U'}{C_{e,V} \cdot T_0} \right) \cdot x_i = m - n \cdot x_i \qquad (5)$$

When the composition of fuel-oxidizer mixture as well as the initial pressure $p_0$ are constant, we can assume that the coefficients $m$ and $n$ (characterized by $\xi$, $r_l$ and $v_l$) are constant. This is true only for a restricted domain of inert gas concentrations. Only when these conditions are fulfilled, the explosion pressure present a linearly dependence on the molar fraction of added inert gas. In this case, a linear dependence of explosion pressure on the molar fraction of added inert gas is observed, as shown by the data from Figure 5.

Even if explosions occur in a spherical vessel, where the explosion is considered to propagate as close as possible to the ideal case, the differences between experimental and ideal adiabatic explosion pressures are significant, as it can be seen from Figure 5. This behaviour shows once more that in the real case, the heat transferred (by convection, conduction or radiation) from the flame front to the vessel walls influences the explosion's propagation. In the present research, the flame was monitored by means of an ionization probe. The ionization probe signal, reaching its peak value at the same time with the peak of the pressure rise rate, can be seen in Figure 6. The observed differences between the experimental and adiabatic values of the explosion pressures can be attributed not only to the heat losses during experiments, but also to the aspect and volume of the explosion vessel, the position of the ignition source or the appearance of turbulences in the gaseous mixture before ignition, which may lead to a decrease in explosion pressure. In the case of an adiabatic process when there are no heat losses, the explosion pressures are influenced only by the initial pressure and temperature and by the composition and the nature of the studied mixture. The differences between the experimental and calculated (adiabatic) explosion pressure are useful to obtain with approximation the amount of lost energy in the course of the combustion process. This was already noted by Kunz in his study on combustion characteristics of $H_2$- and hydrocarbon–air mixtures in closed vessels [36]. Kunz concluded that the larger the differences between the calculated and experimental explosion pressures are, the higher the thermal losses are that occur during the explosion.

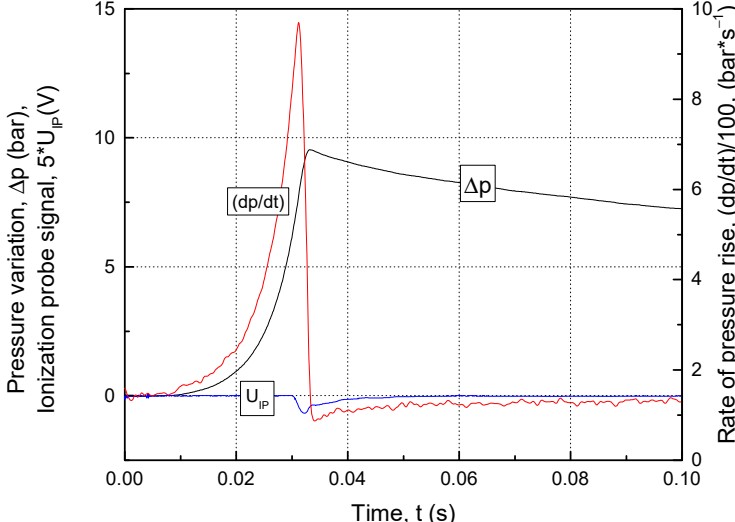

**Figure 6.** Pressure evolution for deflagration of stoichiometric $C_2H_4$-$N_2O$-$N_2$ with 60 vol.% $N_2$, at ambient initial conditions.

The adiabatic explosion pressures have the same behaviour as the experimental explosion pressures does. At a constant amount of nitrogen, higher values of the adiabatic explosion pressures were observed for stoichiometric $C_2H_4$-$N_2O$ mixture compared to

lean $C_2H_4$-$N_2O$ mixture. Keeping the equivalence ratio constant, the adiabatic explosion pressures decreased with an increase in initial pressure.

Set as the zero point (0 s) on an X-axis the moment of ignition. Considering this, the explosion time, $\theta_{max}$, which represents another important combustion parameter, is defined as the explosion duration between the ignition and the moment when the maximum explosion pressure is reached. Adding an inert to a gaseous fuel–oxidizer mixture influences not only the maximum explosion pressure, but also the explosion times. Obviously, with increasing $N_2$ dilution, the explosion duration $\theta_{max}$ increases, as it can be observed from Table 4.

**Table 4.** Time to reach the maximum explosion pressures, $\theta_{max}$ (ms), for ethylene-$N_2O$-$N_2$ mixtures in spherical vessel.

| $\varphi$ | | 1.0 | | | 0.8 | | |
|---|---|---|---|---|---|---|---|
| [$N_2$] (vol.%) | | 40 | 50 | 60 | 40 | 50 | 60 |
| $p_0$ (bar) | | | | | | | |
| 1.5 | | - | 17.1 | 33.3 | - | 21.7 | 43.4 |
| 1.25 | | 10.6 | 17.6 | 32.8 | 13.2 | 22.2 | 43.6 |
| 1 | | 10.7 | 18 | 33.2 | 13.7 | 22.5 | 44.6 |
| 0.75 | | 11.1 | 18.2 | 34.2 | 13.6 | 22.8 | 46 |
| 0.5 | | 11.4 | 18.4 | 34.9 | 13.9 | 23.3 | 51.4 |

It was found that for stoichiometric ethylene-air mixture at $p_0 = 1$ bar, the maximum time to reach $p_{max}$ is $\theta_{max} = 14.6$ ms, higher that for an ethylene-$N_2O$-$N_2$ mixture with the same flame temperature ($\theta_{max} = 10.7$ ms). Lower values of characteristic parameters $\theta_{max}$ were obtained for a stoichiometric $C_2H_4$-$N_2O$-$N_2$ mixture compared to the lean mixture. This behaviour is in accordance with their burning velocities.

The maximum rate of pressure rise represents a safety parameter important for assessing the hazard of a process and, together with the explosion pressure, is used for design of enclosures able to withstand explosions, or of vents used as relief devices for these vessels to avoid the damages caused by explosions occurring in the gaseous phase [37–39]. For explosions of nitrogen-diluted ethylene-$N_2O$ mixtures linear dependences of the form:

$$(dp/dt)_{max} = \alpha + \beta \cdot p_0 \tag{6}$$

were obtained from the graphic representation of the maximum rates of pressure rise versus initial pressures, as can be observed in Figure 7a,b. The coefficients $\alpha$ and $\beta$ depend, on one hand, on the equivalence ratio of studied mixture, and, on the other hand, on the nature and of amount of inert added to the gaseous mixture.

Similar dependences as those given in Figure 7 were reported earlier for fuel–air mixtures such as propane–air [40], LPG–air [41] or propylene–air [42] and for fuel–air–diluent mixtures (e.g., $H_2$–air–steam [43,44], $CH_4$–air–$N_2$, $CH_4$–air–$CO_2$ [45], natural gas–air–$CO_2$ [46]) obtained at sub-atmospheric initial pressures or at initial pressures above 1 bar (with the obvious limitation to deflagrative combustions).

Such diagrams as those depicted in Figure 7 are useful to calculate the maximum rates of pressure rises at any initial pressures different from the ambient one, but within the studied pressure range. This is valid only as long as the explosion propagates as deflagration.

At $p_0 = 0.5$ bar and stoichiometric equivalence ratio of $C_2H_4$-$N_2O$-$N_2$ mixtures, Shen et al. [19] reported $(dp/dt)_{max} = 20$ bar/s for a nitrogen amount of 40 vol.% and $(dp/dt)_{max} = 10$ bar/s for a nitrogen amount of 50 vol.%. As a comparison, the present data are: $(dp/dt)_{max} = 21.0$ bar/s for a nitrogen amount of 40 vol.% and $(dp/dt)_{max} = 14.8$ bar/s for a nitrogen amount of 50 vol.%.

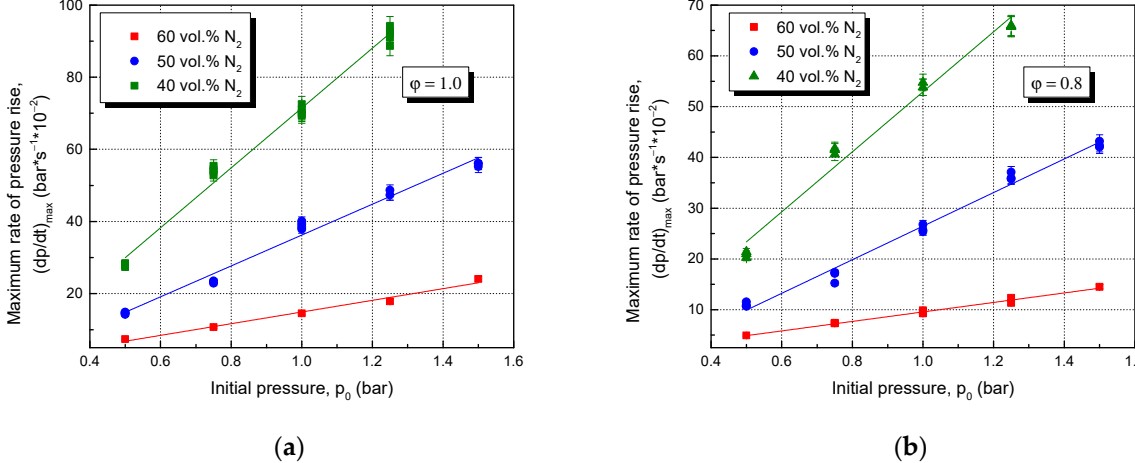

**Figure 7.** Maximum rates of pressure rise for stoichiometric (**a**) and lean (**b**) $C_2H_4$-$N_2O$-$N_2$ mixtures with various amount of $N_2$.

As shown by the above data, the presence of an inert influences the rates of pressure rise. From Figure 7 it is observed that, for constant equivalence ratio, the higher the amount of inert added is, the lower the pressure rise rates are, for both lean and stoichiometric ethylene-$N_2O$ flames. When the amount of $N_2$ is kept constant, it is noticed that the pressure rise rates of lean ethylene-$N_2O$ mixtures are lower compared to the pressure rise rates of stoichiometric mixture. This behaviour was already reported for $CH_4$-$N_2O$-$N_2$ flames [10].

The severity factor is an efficient scaling parameter able to predict the failure effect of enclosures where an explosion occurs. In this study, the severity factors ($K_G$) of $C_2H_4$-$N_2O$-$N_2$ flames were determined by means of the rates of pressure rise obtained from measurements in the spherical vessel with central ignition and are given in Figure 8 versus the initial pressure. The greater the amount of inert is, the lower the severity factors are, a fact observed for both stoichiometric and lean $C_2H_4$-$N_2O$ mixtures when the equivalence ratio is kept constant. As expected, when the inert concentration is kept constant, the higher values of the severity factors are observed for the stoichiometric $C_2H_4$-$N_2O$ mixture compared to lean $C_2H_4$-$N_2O$ mixture.

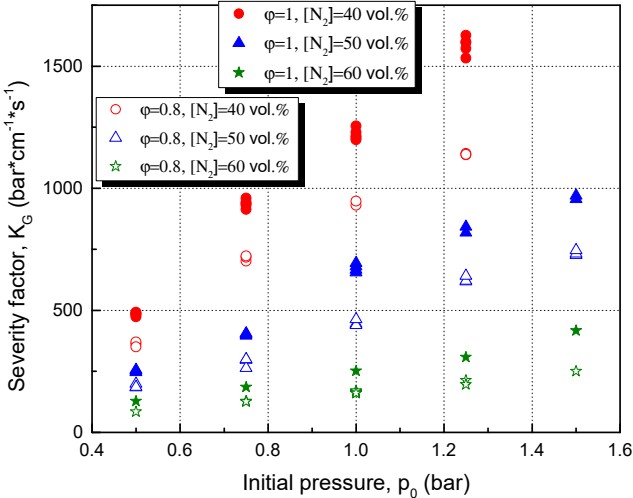

**Figure 8.** Severity factors for $N_2$ diluted lean and stoichiometric $C_2H_4$-$N_2O$-$N_2$ mixtures.

As already reported for $CH_4$-air mixture [47], and as seen from Figure 8, the severity factors are strongly affected by the initial pressure. As expected, the severity factors present

the same linear dependence on the initial pressure as previously observed for the maximum explosion pressures and the rates of pressure rise.

Therefore, the severity of an explosion in fuel–oxidizer mixtures can be reduced by using inert additives which have the role of taking over some of the heat released from the combustion reaction and thereby lowering the temperature in the flame front. The decrease of the flame temperature has as a major effect the decrease of the explosion pressure, accompanied by the decrease of the reaction rate resulting in the decrease of maximum rates of pressure rise and thus of the severity factors.

The maximum explosion pressures and maximum rates of pressure rise of stoichiometric $C_2H_4$-$N_2O$ mixtures diluted with various amounts of $N_2$ or $CO_2$ were studied at various initial pressures using a standard-volume vessel [19]. The authors of this paper concluded that, in the case of the aerospace applications where the dilution degree is small, a $N_2$ dilution contributes better to safety and power efficiency when compared to $CO_2$. On the other hand, in the case of nuclear or other hazardous waste which contains $N_2O$, the nitrogen presents almost the same dilution effect as $CO_2$ due to its presence in large quantities. Therefore, for a better use of these mixtures that present a high explosive risk, specialized scientific studies are always necessary.

## 5. Conclusions

The present paper presents the characteristic parameters of confined explosions ($p_{max}$, $\theta_{max}$ and $(dp/dt)_{max}$) in a spherical vessel with central ignition for lean and stoichiometric $C_2H_4$-$N_2O$ mixtures diluted with various amounts of nitrogen between 40–60 vol.%. The results are examined in correlation with the composition and initial pressure of these flammable mixtures.

The explosion pressures and rates of pressure rise are higher and the times to achieve peak explosion pressures are shorter for nitrogen-diluted $C_2H_4$-$N_2O$ mixtures in comparison to $C_2H_4$-air mixtures with the same flame temperature. The exothermal dissociation of nitrous oxide in $C_2H_4$-$N_2O$-$N_2$ flames is the reason of that behaviour, due to the supplementary heat amount released in fuel–$N_2O$ flames compared to fuel–air flames.

The computed adiabatic explosion pressures are higher in comparison with the measured peak explosion pressures for all flammable mixtures due to the heat losses which take place during the late period of flame propagation.

For all studied flammable mixtures, it was observed that the peak explosion pressures depend linearly on the initial pressures of these mixtures and that the slope as well as the intercept of such dependencies are influenced by the composition of the flammable mixtures.

Linear correlations of the maximum rates of pressure rise and severity factors against the initial pressures were also obtained for all studied $C_2H_4$-$N_2O$-$N_2$ mixtures.

At constant equivalence ratio, the severity factors decrease when the nitrogen amount is increased. At a constant nitrogen amount, the severity factors of stoichiometric $C_2H_4$-$N_2O$-$N_2$ mixture are higher compared to those of lean $C_2H_4$-$N_2O$-$N_2$ mixture.

**Author Contributions:** Conceptualization, V.G., D.R. and C.M.; methodology, C.M. and V.G.; software, D.R.; validation, V.G., M.M., D.R. and C.M.; formal analysis, V.G.; investigation, C.M. and V.G.; resources, C.M. and M.M.; data curation, V.G., M.M., D.R. and C.M.; writing—original draft preparation, C.M. and V.G.; writing—review and editing, V.G., M.M., D.R. and C.M.; visualization, D.R. and C.M; supervision, M.M and D.R..; project administration, C.M.; funding acquisition, C.M. All authors have read and agreed to the published version of the manuscript.

**Funding:** This work was supported by a grant of the Ministry of Research, Innovation and Digitization, CNCS-UEFISCDI, project number PN-III-P4-PCE-2021-0369 within PNCDI III, project number PCE 38/2022.

**Data Availability Statement:** Not applicable.

**Acknowledgments:** This work was partially supported by the Romanian Academy under research project "Dynamics of fast oxidation and decomposition reactions in homogeneous systems" of "Ilie Murgulescu" Institute of Physical Chemistry and by a grant of the Ministry of Research, Innovation and Digitization, CNCS-UEFISCDI, project number PN-III-P4-PCE-2021-0369 within PNCDI III, project number PCE 38/2022.

**Conflicts of Interest:** The authors declare no conflict of interest.

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
