# Peer review of "Dynamics of Pressure Variation in Closed Vessel Explosions of Diluted Fuel/Oxidant Mixtures"

_processes, doi:10.3390/pr10122726_

Round 1

Reviewer 1 Report

The explosion pressure characteristics of diluted fuel/oxidant cylinders are studied in this paper. The paper is well performed, the experiments are reliable, and the data analysis is reasonable. Some innovative research conclusions are obtained, which will promote research in related fields

Author Response

We thank the reviewer for the encouraging words and the appreciation given to our manuscript.

Reviewer 2 Report

Title:

Dynamics of pressure variation in closed vessel explosions of 2 diluted fuel/oxidant mixtures

The main points:

1. Maximum explosion pressures, explosion times, maximum rates of pressure rise, and severity factors. 

2. Various initial pressures (p0 21 = 0.50 – 1.50 bar).

3. Initial pressure and composition influence on pmax, θmax, and (dp/dt)max is discussed.

Point 1:
Lines 105-128: Is the experimental set-up standard? If not, how the self-designed experimental system was validated? Similar question for test conditions.

Point 2:
Is the experimental procedure used standard or self-developed?

Point 3:
Figure 3 Was estimated the relative error over the range of the experimental conditions? Similar questions for Figure 3.

Point 4:
Could the authors write the uncertainty of the computing model results?

Point 5:
The authors do not focus on explaining the scientific meaning of their results. The discussion section is not focused on comparing the results of this study with the literature data.

The paper is within the scope of the journal and has the quality required for publication in the Processes. The present paper is clear and consistent. Therefore, in the current form, after answering mentioned issues, I can recommend it for publication.

Author Response

The main points:

  1. Maximum explosion pressures, explosion times, maximum rates of pressure rise, and severity factors. 
  2. Various initial pressures (p021 = 0.50 – 1.50 bar).
  3. Initial pressure and composition influence on pmax, θmax,and (dp/dt)maxis discussed.

Point 1:

Lines 105-128: Is the experimental set-up standard? If not, how the self-designed experimental system was validated? Similar question for test conditions.

Response:

The experimental set-up was assembled in the Chemical Kinetics Laboratory, Ilie Murgulescu Institute of Physical Chemistry, Bucharest, Romania. It is a self-designed experimental system, validated over the time by comparison with data from literature obtained from measurements in a standard (20 L) spherical vessel and our data are in good agreement with these results. Besides, in a small-scale explosion, the flame front of reactive mixture does not develop significant cellular structures and the buoyancy effects common for larger vessels are reduced. In addition, in large-scale enclosures, there is also the possibility of flame acceleration and transition to a turbulent combustion regime. Therefore, adequate safety protective measures can be formulated only on the basis of systematic studies concerning the characteristic indices of deflagrations taking place in enclosures.

The studied C2H4/N2O ratios, as shown by these equivalence ratios, were chosen as representative since they afford the complete fuel consumption and generally ensures diffusion stability of the flame front. Given the fact that, at elevated pressures and temperatures, N2O it can easily decompose to oxygen and nitrogen, these values of the initial temperature and pressure were chosen for concerns of laboratory operation safety.

Point 2:

Is the experimental procedure used standard or self-developed?

Response:

The experimental procedure is in accordance with the standard test procedure and consisted of evacuating the combustion vessel down to 0.1 mbar; the fuel–air mixture was then introduced, allowed to become quiescent and ignited. Minimum three experiments were performed for each initial condition of explosive mixture. For a few systems several sets of 15 experiments were conducted in identical conditions, in the spherical vessel with central ignition to ensure the accuracy and repeatability of the data.

The average error in measured explosion pressure was 2.5%.

Point 3:

Figure 3 Was estimated the relative error over the range of the experimental conditions? Similar questions for Figure 3.

Response:

The uncertainties involved in the present study can be due to mixture preparation, data acquisition and data processing. The uncertainty due to the mixture preparation depends strongly on the accuracy of the pressure gauge manometers. In this study, very accurate pressure gauge manometers were used; therefore, the uncertainties due to the mixtures preparation can be neglected. The uncertainty during an experiment can be also attributed to the initial conditions: temperature and pressure. During all experiments, the initial temperature of the mixtures was considered the room temperature (298 K). However, an estimated error of 2 K (about 0.7%) due to the temperature variation from one day to another can be considered a possible factor that may affect the overall precision of measurements. In addition to the errors that may be due to the initial temperature, other errors which may arise correspond to the accuracy of the pressure transducer and ignition sources. According to the piezoelectric pressure transducer manufacturer, the error given by this device is less or equal to 0.1%. Considering all of these possible sources of experimental errors, a total standard deviation of 2.5% was taking into account during this study.

Point 4:
Could the authors write the uncertainty of the computing model results?

Response:

The uncertainties of the computed results may arise from the erroneous data taken from international databases, i.e. the polynomials of heat capacities as functions of temperature. We cannot estimate these errors and we hope they do not influence too much our results.

COSILAB package is a tool for simulating a variety of laminar flames including unstrained (such as premixed freely propagating flames, premixed burner-stabilized flames) or strained premixed flames (such as diffusion flames, partially premixed cylindrically or spherically). This software allows solving complex chemical kinetics problems which involve thousands of reactions. The runs can be made by varying some physical parameters such as equivalence ratio, temperature, pressure, and strain rate. The COSILAB capabilities allow a detailed study on a complex chemical reaction considering intermediate compounds, trace compounds and pollutants.

0D COSILAB package is based on a general algorithm meant to compute the equilibrium composition of products for fuel–oxidant gaseous mixtures using the thermodynamic criteria of chemical equilibrium. Using this package and the recommended computing protocol, we can determine the mole numbers of all species and the temperature of the burned gas in the isobaric combustion by computations at constant p and H. The same is valid for the isochoric combustion, when the runs are made at constant V and U.

Point 5:

The authors do not focus on explaining the scientific meaning of their results. The discussion section is not focused on comparing the results of this study with the literature data.

Response:

The study of hydrocarbon-nitrous oxide combustion can provide valuable information on the fundamental chemical kinetics relevant to complex oxidation systems involving nitrogen. Researchers have focused their interest mostly for studying the explosion characteristics of other fuel-N2O flames than of ethylene-N2O flames. Ethylene-N2O mixtures are used in aerospace applications as a promising green propellant for hydrazine replacement. Despite their advantages in comparison with hydrazine (a relatively low cost, a high vapor pressure, a low toxicity, a good miscibility), ethylene-nitrous oxide mixtures are flammable mixtures. Thus, it fully justifies the interest on a comprehensive study of their explosivity to help formulate answers related to three categories of problems: prediction, prevention and protection.

As it is mentioned in the manuscript, there is a lack of data referring to ethylene combustion in the presence of nitrous oxide, therefore our results (registered in a spherical vessel) could only be compared with data obtained by Shen et al. from measurements in a spherical vessel at sub-atmospheric pressures.

[19]. Shen X, Zhang N, Shi X, Cheng X. Experimental studies on pressure dynamics of C2H4/N2O mixtures explosion with dilution. Appl. Therm. Eng. 2019; 147:74–80.

The paper is within the scope of the journal and has the quality required for publication in the Processes. The present paper is clear and consistent. Therefore, in the current form, after answering mentioned issues, I can recommend it for publication.

Response:

The authors hope that the reviewer will find that answers to all questions and comments have been provided.

Reviewer 3 Report

Text of the article please adjust as indicated in the instructions

Use the Microsoft Word template to prepare your manuscript; Submission Checklist.

Specific comments are in the attachment.

Author Response

Type of paper: Article

Dynamics of pressure variation in closed vessel explosions of diluted fuel/oxidant mixtures

Venera Giurcan, Domnina Razus, Maria Mitu and Codina Movileanu

Thank you for developing your interesting article on experiments.

I have the following comments.

Specifically:

Text of the article please adjust as indicated in the instructions

Use the Microsoft Word template to prepare your manuscript; Submission Checklist

  1. Introduction
  2. Materials and Methods

…….

  1. Results

3.1. Subsection

3.1.1. Subsubsection

Response:

The text article was adjusted as indicated in the instructions.

In the text of the entire article, please use physical units uniformly. For example g∙m-3, MPa∙m∙s-1 bar∙m∙s-1, MPa∙s-1, m-2∙s-2, etc.

Response:

The physical units were uniformized.

In figures 2, 3, 4, 5, 6, 7 and 8, complete the description of the legend in the x-axis and in the y-axis with a word. pressure and in parentheses indicate the physical unit (bar)

Response:

The legend in the x-axis and in the y-axis has the format usually used so that it is explicit and and, therefore, we consider that it is not necessary to aglomerate the graph.

Figure 6 - edit the description of the legend on the left side - rotate the text

Response:

Figure 6 was edited.

Edit all tables in the article according to the pattern

Lines – 265, center the equation

Lines – 278, center the equation

Lines – 287, center the equation

Lines – 339, center the equation

Please use black numbers to cite the publication.

Response:

The tables and equation were edited.

The black numbers were used to cite the publication.

Round 2

Reviewer 3 Report

Thank you for making adjustments to the article according to the comments.

Pleas you, Complete the description of the tables in the images.

In figures 2, 3, 4, 5, 6, 7 and 8, complete the description of the legend in the x-axis and in the y-axis with a word. For example Figure 2, on the x-axis you list t (s). Write in a word. time (s) Similarly, write ∆p (bar) on the y axis.

Author Response

The changes in Figures 2-8 were made.